# HyperGCN: A New Method of Training Graph Convolutional Networks on Hypergraphs

**Naganand Yadati**
Indian Institute of Science, Bangalore
y.naganand@gmail.com

**Madhav Nimishakavi**
Indian Institute of Science, Bangalore
cse.madhav@gmail.com

**Prateek Yadav**
Indian Institute of Science, Bangalore
ugprateek@gmail.com

**Vikram Nitin** *
Birla Institute of Technology and Science, Pilani
vikramnitin9@gmail.com

**Anand Louis**
Indian Institute of Science, Bangalore
anandl@iisc.ac.in

**Partha Talukdar**
Indian Institute of Science, Bangalore
partha@talukdar.net

## Abstract

In many real-world networks such as co-authorship, co-citation, etc., relationships are complex and go beyond pairwise associations. Hypergraphs provide a flexible and natural modeling tool to model such complex relationships. The obvious existence of such complex relationships in many real-world networks naturally motivates the problem of learning with hypergraphs. A popular learning paradigm is hypergraph-based semi-supervised learning (SSL) where the goal is to assign labels to initially unlabelled vertices in a hypergraph. Motivated by the fact that a graph convolutional network (GCN) has been effective for graph-based SSL, we propose HyperGCN, a novel way of training a GCN for SSL on hypergraphs based on tools from sepctral theory of hypergraphs. We demonstrate HyperGCN's effectiveness through detailed experimentation on real-world hypergraphs for SSL and combinatorial optimisation and analyse when it is going to be more effective than state-of-the art baselines. We have made the source code available.

## 1 Introduction

In many real-world network datasets such as co-authorship, co-citation, email communication, etc., relationships are complex and go beyond pairwise associations. Hypergraphs provide a flexible and natural modeling tool to model such complex relationships. For example, in a co-authorship network an author (hyperedge) can be a co-author of more than two documents (vertices).

The obvious existence of such complex relationships in many real-world networks naturaly motivates the problem of learning with hypergraphs [52, 22, 49, 17]. A popular learning paradigm is graph-based / hypergraph-based semi-supervised learning (SSL) where the goal is to assign labels to initially unlabelled vertices in a graph / hypergraph [10, 54, 42]. While many techniques have used explicit Laplacian regularisation in the objective [51, 53, 11, 48], the state-of-the-art neural methods encode the graph / hypergraph structure $G = (V, E)$ *implicitly* via a neural network $f(G, X)$[25, 3, 17] ($X$ contains the initial features on the vertices for example, text attributes for documents).

| Model↓    Metric → | Training time | Density | Training time (DBLP) | Training time (Pubmed) |
|---|---|---|---|---|
| HGNN | 170s | 337 | 0.115s | 0.019s |
| FastHyperGCN | **143**s | **352** | **0.035**s | **0.016**s |

Table 1: average training time of an epoch (lower is better)

While explicit Laplacian regularisation assumes similarity among vertices in each edge / hyperedge, implicit regularisation of graph convolutional networks (GCNs) [25] avoids this restriction and enables application to a broader range of problems in combinatorial optimisation [19, 26, 38, 31], computer vision [12, 37], natural language processing [44, 34], etc. In this work, we propose, HyperGCN, a novel training scheme for a GCN on hypergraph and show its effectiveness not only in SSL where hyperedges encode similarity but also in combinatorial optimisation where hyperedges do not encode similarity. Combinatorial optimisation on hypergraphs has recently been highlighted as crucial for real-world network analysis [2, 36].

Methodologically, HyperGCN approximates each hyperedge of the hypergraph by a set of pairwise edges connecting the vertices of the hyperedge and treats the learning problem as a graph learning problem on the approximation. While the state-of-the-art hypergraph neural networks (HGNN) [17] approximates each hyperedge by a clique and hence requires ${}^sC_2$ (quadratic number of) edges for each hyperedge of size $s$, our method, i.e. HyperGCN, requires a linear number of edges (i.e. $O(s)$) for each hyperedge. The advantage of this linear approximation is evident in Table 1 where a faster variant of our method has lower training time on synthetic data (with higher density as well) for densest $k$-subhypergraph and SSL on real-world hypergraphs (DBLP and Pubmed). In summary, we make the following contributions:

- We propose HyperGCN, a new method of training a GCN on hypergraph using tools from spectral theory of hypergraphs and introduce FastHyperGCN, its faster variant (Section 4).
- We apply our methods to the problems of SSL and combinatorial optimisation on real-world hypergraphs. Through detailed experimentation, we demonstrate their effectiveness compared to the state-of-the art HGNN [17] and other baselines (Sections 5, and 7).
- We thoroughly discuss when we prefer our methods to HGNN (Sections 6, and 8).

While the motivation of our methods is based on similarity of vertices in a hyperedge, we show they can be effectively used for combinatorial optimisation where hyperedges do not encode similarity.

## 2  Related work

In this section, we discuss related work and then the background in the next section.

**Deep learning on graphs**: *Geometric deep learning* [5] is an umbrella phrase for emerging techniques attempting to generalise (structured) deep neural network models to non-Euclidean domains such as graphs and manifolds. Graph convolutional network (GCN) [25] defines the convolution using a simple linear function of the graph Laplacian and is shown to be effective on semi-supervised classification on attributed graphs. The reader is referred to a comprehensive literature review [5] and extensive surveys [20, 4] on this topic of deep learning on graphs.

**Learning on hypergraphs**: The clique expansion of a hypergraph was introduced in a seminal work [52] and has become a popular approach for learning on hypergraph-structured data [39, 16, 50, 43]. Hypergraph neural networks [17] and their variants [23, 24] use the clique expansion to extend GCNs for hypergraphs. Powerset convolutional networks [47] utilise tools from signal processing to define convolution on set functions. Another line of work uses mathematically appealing tensor methods [40, 6], but they are limited to uniform hypergraphs.

**Spectral Theory of Hypergraphs**: The clique expansion of a hypergraph essentially models it as a graph by converting each hyperedge to a clique subgraph [1]. It has been well established that this approximation causes distortion, fails to utilise higher-order relationships in the data and leads to unreliable learning performance for clustering, SSL, active learning, etc. [28, 13]. A simple yet effective way to overcome the limitations is to introduce hyperedge-dependent vertex weights [14].

Researchers have fully utilised the hypergraph structure also through non-linear Laplacian operators [22, 32, 8]. It has been shown that these operators enable Cheeger-type inequality for hypergraphs,

relating the second smallest eigenvalue of the operator to hypergraph expansion [32, 8]. One such Laplacian is derived from the notion of total variation on hypergraphs (Lovasz extension of the hypergraph cut) which considers the maximally disparate vertices in each hyperedge [22]. Recent developements have extended these non-linear operators to several different settings:

- directed hypergraphs (idea is to consider supremum in tail, infimum in head) [49, 9].

- submodular hypergraphs (different submodular weights for different hyperedge cuts) [30] and submodular function minimisation (generalises hypergraph SSL objective) [27, 29].

- Laplacian that considers all vertices in each hyperedge (includes the vertices other than the maximally disparate ones in each hyperedge) [7].

**Graph-based SSL**: Researchers have shown that using unlabelled data in training can improve learning accuracy significantly. This topic is so popular that it has influential books [10, 54, 42].

**Graph neural networks for combinatorial optimisation**: Graph-based deep models have recently been shown to be effective as learning-based approaches for NP-hard problems such as maximal independent set, minimum vertex cover, etc. [31], the decision version of the traveling salesman problem [38], graph colouring [26], and clique optimisation [19].

## 3 Background: Graph convolutional network

Let $\mathcal{G} = (\mathcal{V}, \mathcal{E})$, with $N = |\mathcal{V}|$, be a simple undirected graph with adjacency $A \in \mathbb{R}^{N \times N}$, and data matrix $X \in \mathbb{R}^{N \times p}$. which has $p$-dimensional real-valued vector representations for each node $v \in \mathcal{V}$.

The basic formulation of graph convolution [25] stems from the convolution theorem [33] and it can be shown that the convolution $C$ of a real-valued graph signal $S \in \mathbb{R}^N$ and a filter signal $F \in \mathbb{R}^N$ is approximately $C \approx (w_0 + w_1 \tilde{L})S$ where $w_0$ and $w_1$ are learned weights, and $\tilde{L} = \frac{2L}{\lambda_N} - I$ is the scaled graph Laplacian, $\lambda_N$ is the largest eigenvalue of the symmetrically-normalised graph Laplacian $L = I - D^{-\frac{1}{2}} A D^{-\frac{1}{2}}$ where $D = \text{diag}(d_1, \cdots, d_N)$ is the diagonal degree matrix with elements $d_i = \sum_{j=1, j \neq i}^{N} A_{ji}$. The filter $F$ depends on the structure of the graph (the graph Laplacian $L$). The detailed derivation from the convolution theorem uses existing tools from graph signal processing [41, 21, 5] and is provided in the supplementary material. The key point here is that the convolution of two graph signals is a *linear function* of the graph Laplacian $L$.

Table 2: Summary of symbols used in the paper.

| Symbol | Description | Symbol | Description |
|---|---|---|---|
| $\mathcal{G} = (\mathcal{V}, \mathcal{E})$ | an undirected simple graph | $\mathcal{H} = (V, E)$ | an undirected hypergraph |
| $\mathcal{V}$ | set of nodes | $V$ | set of hypernodes |
| $\mathcal{E}$ | set of edges | $E$ | set of hyperedges |
| $N = |\mathcal{V}|$ | number of nodes | $n = |V|$ | number of hypernodes |
| $L$ | graph Laplacian | $\mathbb{L}$ | hypergraph Laplacian |
| $A$ | graph adjacency matrix | $H$ | hypergraph incidence matrix |

The graph convolution for $p$ different graph signals contained in the data matrix $X \in \mathbb{R}^{N \times p}$ with learned weights $\Theta \in \mathbb{R}^{p \times r}$ with $r$ hidden units is $\bar{A} X \Theta$, $\bar{A} = \tilde{D}^{-\frac{1}{2}} \tilde{A} \tilde{D}^{-\frac{1}{2}}$, $\tilde{A} = A + I$, and $\tilde{D}_{ii} = \sum_{j=1}^{N} \tilde{A}_{ij}$. The proof involves a renormalisation trick [25] and is in the supplementary.

**GCN [25]** The forward model for a simple two-layer GCN takes the following simple form:

$$Z = f_{GCN}(X, A) = \text{softmax}\left( \bar{A} \ \text{ReLU}\left( \bar{A} X \Theta^{(1)} \right) \Theta^{(2)} \right), \tag{1}$$

where $\Theta^{(1)} \in \mathbb{R}^{p \times h}$ is an input-to-hidden weight matrix for a hidden layer with $h$ hidden units and $\Theta^{(2)} \in \mathbb{R}^{h \times r}$ is a hidden-to-output weight matrix. The softmax activation function is defined as $\text{softmax}(x_i) = \frac{\exp(x_i)}{\sum_j \exp(x_j)}$ and applied row-wise.

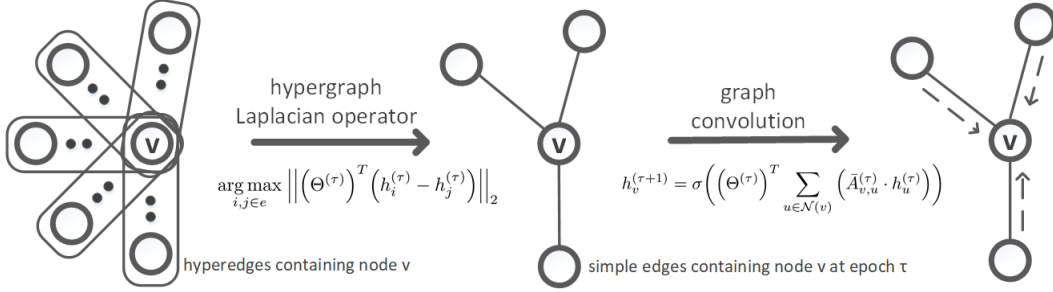

Figure 1: Graph convolution on a hypernode $v$ using HyperGCN.

**GCN training for SSL**: For multi-class, classification with $q$ classes, we minimise cross-entropy,

$$\mathcal{L} = -\sum_{i \in \mathcal{V}_L} \sum_{j=1}^{q} Y_{ij} \ln Z_{ij}, \tag{2}$$

over the set of labelled examples $\mathcal{V}_L$. Weights $\Theta^{(1)}$ and $\Theta^{(2)}$ are trained using gradient descent.

A summary of the notations used throughout our work is shown in Table 2.

## 4   HyperGCN: Hypergraph Convolutional Network

We consider semi-supervised hypernode classification on an undirected hypergraph $\mathcal{H} = (V, E)$ with $|V| = n$, $|E| = m$ and a small set $V_L$ of labelled hypernodes. Each hypernode $v \in V = \{1, \cdots, n\}$ is also associated with a feature vector $x_v \in \mathbb{R}^p$ of dimension $p$ given by $X \in \mathbb{R}^{n \times p}$. The task is to predict the labels of all the unlabelled hypernodes, that is, all the hypernodes in the set $V \setminus V_L$.

**Overview**: The crucial working principle here is that the hypernodes in the same hyperedge are similar and hence are likely to share the same label [49]. Suppose we use $\{h_v : v \in V\}$ to denote some representation of the hypernodes in $V$, then, for any $e \in E$, the function $\max_{i,j \in e} ||h_i - h_j||^2$ will be "small" only if vectors corresponding to the hypernodes in $e$ are "close" to each other. Therefore, $\sum_{e \in E} \max_{i,j \in e} ||h_i - h_j||^2$ as a regulariser is likely to achieve the objective of the hypernodes in the same hyperedge having similar representations. However, instead of using it as an explicit regulariser, we can achieve the same goal by using GCN over an appropriately defined Laplacian of the hypergraph. In other words, we use the notion of *hypergraph Laplacian* as an implicit regulariser which achieves this objective.

A hypergraph Laplacian with the same underlying motivation as stated above was proposed in prior works [8, 32]. We present this Laplacian first. Then we run GCN over the simple graph associated with this hypergraph Laplacian. We call the resulting method 1-HyperGCN (as each hyperedge is approximated by exactly one pairwise edge). One epoch of 1-HyperGCN is shown in figure 1

### 4.1   Hypergraph Laplacian

As explained before, the key element for a GCN is the graph Laplacian of the given graph $\mathcal{G}$. Thus, in order to develop a GCN-based SSL method for hypergraphs, we first need to define a Laplacian for hypergraphs. One such way [8] (see also [32]) is a non-linear function $\mathbb{L} : \mathbb{R}^n \to \mathbb{R}^n$ (the Laplacian matrix for graphs can be viewed as a linear function $L : \mathbb{R}^n \to \mathbb{R}^n$).

**Definition 1 (Hypergraph Laplacian [8, 32]$^2$)** *Given a real-valued signal $S \in \mathbb{R}^n$ defined on the hypernodes, $\mathbb{L}(S)$ is computed as follows.*

    *1. For each hyperedge $e \in E$, let $(i_e, j_e) := argmax_{i,j \in e} |S_i - S_j|$, breaking ties randomly$^2$.*

---

$^2$The problem of breaking ties in choosing $i_e$ (resp. $j_e$) is a non-trivial problem as shown in [8]. Breaking ties randomly was proposed in [32], but [8] showed that this might not work for all applications (see [8] for more details). [8] gave a way to break ties, and gave a proof of correctness for their tie-breaking rule for the problems they studied. We chose to break ties randomly because of its simplicity and its efficiency.

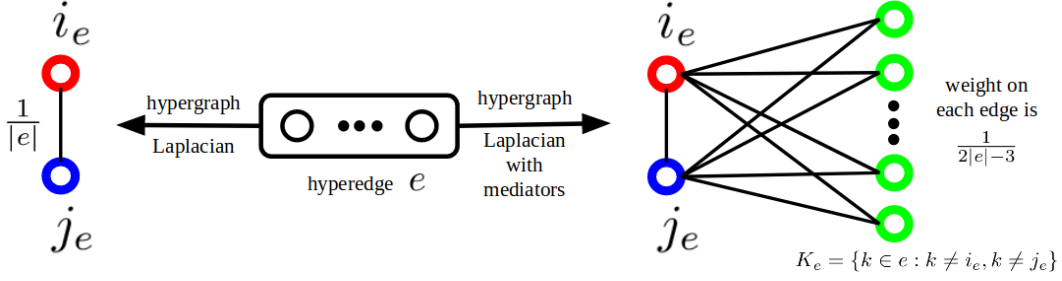

Figure 2: Hypergraph Laplacian [8] vs. the generalised hypergraph Laplacian with mediators [7]. Our approach requires at most a linear number of edges (1 and $2|e| - 3$ respectively) while HGNN [17] requires a quadratic number of edges for each hyperedge.

2. *A weighted graph $G_S$ on the vertex set $V$ is constructed by adding edges $\{\{i_e, j_e\} : e \in E\}$ with weights $w(\{i_e, j_e\}) := w(e)$ to $G_S$, where $w(e)$ is the weight of the hyperedge $e$. Let $A_S$ denote the weighted adjacency matrix of the graph $G_S$.*

3. *The symmetrically normalised hypergraph Laplacian is $\mathbb{L}(S) := (I - D^{-\frac{1}{2}} A_S D^{-\frac{1}{2}})S$*

### 4.2   1-**HyperGCN**

By following the Laplacian construction steps outlined in Section 4.1, we end up with the simple graph $G_S$ with normalized adjacency matrix $\bar{A}_S$. We now perform GCN over this simple graph $G_S$. The graph convolution operation in Equation (1), when applied to a hypernode $v \in V$ in $G_S$, in the neural message-passing framework [18] is $h_v^{(\tau+1)} = \sigma\left((\Theta^{(\tau)})^T \sum_{u \in \mathcal{N}(v)} ([\bar{A}_S^{(\tau)}]_{v,u} \cdot h_u^{(\tau)})\right)$.

Here, $\tau$ is epoch number, $h_v^{(\tau+1)}$ is the new hidden layer representation of node $v$, $\sigma$ is a non-linear activation function, $\Theta$ is a matrix of learned weights, $\mathcal{N}(u)$ is the set of neighbours of $v$, $[\bar{A}_S^{(\tau)}]_{v,u}$ is the weight on the edge $\{v, u\}$ after normalisation, and $h_u^{(\tau)}$ is the previous hidden layer representation of the neighbour $u$. We note that along with the embeddings of the hypernodes, the adjacency matrix is also re-estimated in each epoch.

Figure 1 shows a hypernode $v$ with five hyperedges incident on it. We consider exactly one representative simple edge for each hyperedge $e \in E$ given by $(i_e, j_e)$ where $(i_e, j_e) = \arg\max_{i,j \in e} ||(\Theta^{(\tau)})^T (h_i^{(\tau)} - h_j^{(\tau)})||_2$ for epoch $\tau$. Because of this consideration, the hypernode $v$ may not be a part of all representative simple edges (only three shown in figure). We then use traditional Graph Convolution Operation on $v$ considering only the simple edges incident on it. Note that we apply the operation on each hypernode $v \in V$ in each epoch $\tau$ of training until convergence.

**Connection to total variation on hypergraphs**: Our 1-HyperGCN model can be seen as performing implicit regularisation based on the total variation on hypergraphs [22]. In that prior work, explicit regularisation and only the hypergraph structure is used for hypernode classification in the SSL setting. HyperGCN, on the other hand, can use both the hypergraph structure and also exploit any available features on the hypernodes, e.g., text attributes for documents.

### 4.3   **HyperGCN: Enhancing** 1-**HyperGCN with mediators**

One peculiar aspect of the hypergraph Laplacian discussed is that each hyperedge $e$ is represented by a single pairwise simple edge $\{i_e, j_e\}$ (with this simple edge potentially changing from epoch to epoch). This hypergraph Laplacian ignores the hypernodes in $K_e := \{k \in e : k \neq i_e, k \neq j_e\}$ in the given epoch. Recently, it has been shown that a generalised hypergraph Laplacian in which the hypernodes in $K_e$ act as "mediators" [7] satisfies all the properties satisfied by the above Laplacian given by [8]. The two Laplacians are pictorially compared in Figure 2. Note that if the hyperedge is of size 2, we connect $i_e$ and $j_e$ with an edge. We also run a GCN on the simple graph associated with the hypergraph Laplacian with mediators [7] (right in Figure 2). It has been suggested that the

Table 3: Real-world hypergraph datasets used in our work. Distribution of hyperedge sizes is not symmetric either side of the mean and has a strong positive skewness.

| | DBLP (co-authorship) | Pubmed (co-citation) | Cora (co-authorship) | Cora (co-citation) | Citeseer (co-citation) |
|---|---|---|---|---|---|
| # hypernodes, $|V|$ | 43413 | 19717 | 2708 | 2708 | 3312 |
| # hyperedges, $|E|$ | 22535 | 7963 | 1072 | 1579 | 1079 |
| **avg. hyperedge size** | **$4.7 \pm 6.1$** | **$4.3 \pm 5.7$** | **$4.2 \pm 4.1$** | **$3.0 \pm 1.1$** | **$3.2 \pm 2.0$** |
| # features, $d$ | 1425 | 500 | 1433 | 1433 | 3703 |
| # classes, $q$ | 6 | 3 | 7 | 7 | 6 |
| label rate, $|V_L|/|V|$ | 0.040 | 0.008 | 0.052 | 0.052 | 0.042 |

weights on the edges for each hyperedge in the hypergraph Laplacian (with mediators) sum to 1 [7]. We chose each weight to be $\frac{1}{2|e|-3}$ as there are $2|e| - 3$ edges for a hyperedge $e$.

## 4.4 FastHyperGCN

We use just the initial features $X$ (without the weights) to construct the hypergraph Laplacian matrix (with mediators) and we call this method FastHyperGCN. Because the matrix is computed only once before training (and not in each epoch), the training time of FastHyperGCN is much less than that of other methods. Please see the supplementrary for all the algorithms.

# 5 Experiments for semi-supervised learning

We conducted experiments not only on real-world datasets but also on categorical data (results in supplementary) which are a standard practice in hypergraph-based learning [52, 22, 49, 30, 29, 27].

## 5.1 Baselines

We compared HyperGCN, 1-HyperGCN and FastHyperGCN against the following baselines:

- **Hypergraph neural networks (HGNN) [17]** uses the clique expansion [52, 1] to approximate the hypergraph. Each hyperedge of size $s$ is approximated by an $s$-clique.

- **Multi-layer perceptron (MLP)** treats each instance (hypernode) as an independent and identically distributed (i.i.d) instance. In other words, $A = I$ in equation 1. We note that this baseline does not use the hypergraph structure to make predictions.

- **Multi-layer perceptron + explicit hypergraph Laplacian regularisation (MLP + HLR)**: regularises the MLP by training it with the loss given by $\mathcal{L} = \mathcal{L}_0 + \lambda \mathcal{L}_{reg}$ and uses the hypergraph Laplacian with mediators for explicit Laplacian regularisation $L_{reg}$. We used $10\%$ of the test set used for all the above models for this baseline to get an optimal $\lambda$.

- **Confidence Interval-based method (CI) [49]** uses a subgradient-based method [49]. We note that this method has consistently been shown to be superior to the primal dual hybrid gradient (PDHG) of [22] and also [52]. Hence, we did not use these other previous methods as baselines, and directly compared HyperGCN against CI.

The task for each dataset is to predict the topic to which a document belongs (multi-class classification). Statistics are summarised in Table 3. For more details about datasets, please refer to the supplementary. We trained all methods for 200 epochs and used the same hyperparameters of a prior work [25]. We report the mean test error and standard deviation over 100 different train-test splits. We sampled sets of same sizes of labelled hypernodes from each class to have a balanced train split.

# 6 Analysis of results

The results on real-world datasets are shown in Table 4. Firstly we note that HyperGCN is superior to 1-HyperGCN. This is expected as all the vertices in a hyperedge participate in the hypergraph Laplacian in HyperGCN while only two in 1-HyperGCN. Interestingly, we note that FastHyperGCN is superior to 1-HyperGCN. This, we believe is because of the large hyperedges (size more than 4)

Table 4: Results of SSL experiments. We report mean test error $\pm$ standard deviation (lower is better) over 100 train-test splits. Please refer to section 5 for details.

| Data | Method | DBLP co-authorship | Pubmed co-citation | Cora co-authorship | Cora co-citation | Citeseer co-citation |
|------|--------|--------------------|--------------------|--------------------|--------------------|----------------------|
| $\mathcal{H}$ | CI | $54.81 \pm 0.9$ | $52.96 \pm 0.8$ | $55.45 \pm 0.6$ | $64.40 \pm 0.8$ | $70.37 \pm 0.3$ |
| $\mathbf{X}$ | MLP | $37.77 \pm 2.0$ | $30.70 \pm 1.6$ | $41.25 \pm 1.9$ | $42.14 \pm 1.8$ | $41.12 \pm 1.7$ |
| $\mathcal{H}, \mathbf{X}$ | MLP + HLR | $30.42 \pm 2.1$ | $30.18 \pm 1.5$ | $34.87 \pm 1.8$ | $36.98 \pm 1.8$ | $37.75 \pm 1.6$ |
| $\mathcal{H}, \mathbf{X}$ | HGNN | $25.65 \pm 2.1$ | $29.41 \pm 1.5$ | $31.90 \pm 1.9$ | $\mathbf{32.41 \pm 1.8}$ | $\mathbf{37.40 \pm 1.6}$ |
| $\mathcal{H}, \mathbf{X}$ | 1-HyperGCN | $33.87 \pm 2.4$ | $30.08 \pm 1.5$ | $36.22 \pm 2.2$ | $34.45 \pm 2.1$ | $38.87 \pm 1.9$ |
| $\mathcal{H}, \mathbf{X}$ | FastHyperGCN | $27.34 \pm 2.1$ | $29.48 \pm 1.6$ | $32.54 \pm 1.8$ | $\mathbf{32.43 \pm 1.8}$ | $37.42 \pm 1.7$ |
| $\mathcal{H}, \mathbf{X}$ | HyperGCN | $\mathbf{24.09 \pm 2.0}$ | $\mathbf{25.56 \pm 1.6}$ | $\mathbf{30.08 \pm 1.8}$ | $32.37 \pm 1.7$ | $37.35 \pm 1.6$ |

Table 5: Results (lower is better) on sythetic data and a subset of DBLP showing that our methods are more effective for noisy hyperedges. $\eta$ is no. of hypernodes of one class divided by that of the other in noisy hyperedges. Best result is in bold and second best is underlined. Please see Section 6.

| Method | $\eta = 0.75$ | $\eta = 0.70$ | $\eta = 0.65$ | $\eta = 0.60$ | $\eta = 0.55$ | $\eta = 0.50$ | sDBLP |
|--------|---------------|---------------|---------------|---------------|---------------|---------------|-------|
| HGNN | $\mathbf{15.92 \pm 2.4}$ | $\mathbf{24.89 \pm 2.2}$ | $\mathbf{31.32 \pm 1.9}$ | $39.13 \pm 1.78$ | $42.23 \pm 1.9$ | $44.25 \pm 1.8$ | $45.27 \pm 2.4$ |
| FastHyperGCN | $28.86 \pm 2.6$ | $31.56 \pm 2.7$ | $33.78 \pm 2.1$ | $\underline{33.89 \pm 2.0}$ | $\mathbf{34.56 \pm 2.2}$ | $\mathbf{35.65 \pm 2.1}$ | $\underline{41.79 \pm 2.8}$ |
| HyperGCN | $\underline{22.44 \pm 2.0}$ | $\underline{29.33 \pm 2.2}$ | $\underline{33.41 \pm 1.9}$ | $\mathbf{33.67 \pm 1.9}$ | $\underline{35.05 \pm 2.0}$ | $\underline{37.89 \pm 1.9}$ | $\mathbf{41.64 \pm 2.6}$ |

present in all the datasets. FastHyperGCN uses all the mediators while 1-HyperGCN uses only two vertices. We now attempt to explain them.

**Proposition 1:** Given a hypergraph $\mathcal{H} = (V, E)$ with $E \subseteq 2^V - \cup_{v \in V}\{v\}$ and signals on the vertices $S : V \to R^d$, let, for each hyperedge $e \in E$, $(i_e, j_e) := \arg\max_{i,j \in e} ||S_i - S_j||_2$ and $K_e := \{v \in e : v \neq i_e, v \neq j_e\}$. Define

- $E_c := \bigcup_{e \in E} \left\{ \{u, v\} : u \in e, v \in e, u \neq v \right\}$

- $w_c\left(\{u, v\}\right) := \sum_{e \in E} \mathbb{1}_{\{u,v\} \in E_c} \cdot \mathbb{1}_{u \in e} \cdot \mathbb{1}_{v \in e}\left(\frac{2}{|e| \cdot (|e|-1)}\right),$

- $E_m(S) := \bigcup_{e \in E} \{i_e, j_e\} \bigcup_{e \in E, |e| \geq 3} \left\{ \{u, v\} : u \in \{i_e, j_e\}, v \in K_e \right\}$

- $w_m\left(S, \{u, v\}\right) := \sum_{e \in E} \mathbb{1}_{\{u,v\} \in E_m(S)} \cdot \mathbb{1}_{u \in e} \cdot \mathbb{1}_{v \in e}\left(\frac{1}{2|e|-3}\right),$

so that $G_c = (V, E_c, w_c)$ and $G_m(S) = (V, E_m(S), w_m(S))$ are the normalised clique expansion, i.e., graph of HGNN and mediator expansion, i.e., graph of HyperGCN/FastHyperGCN respectively. *A sufficient condition for $G_c = G_m(S), \forall S$ is $\max_{e \in E} |e| = 3$.*

**Proof:** Observe that we consider hypergraphs in which the size of each hyperedge is at least 2. It follows from definitions that $|E_c| = \sum_{e \in E} {}^{|e|}C_2$ and $|E_m| = \sum_{e \in E}\left(2|e| - 3\right)$. Clealy, a sufficient condition is when each hyperedge is approximated by the same subgraph in both the expansions. In other words the condition is $\frac{|e| \cdot (|e|-1)}{2} = 2|e| - 3$ for each $e \in E$. Solving the resulting quadratic eqution $x^2 - 5x + 6 = 0$ gives us $(x - 2)(x - 3) = 0$. Hence, $|e| = 2$ or $|e| = 3$ for each $e \in E$. $\square$

**Comparable performance on Cora and Citeseer co-citation**
We note that HGNN is the most competitive baseline. Also $S = X$ for FastHyperGCN and $S = H\Theta$

for HyperGCN. The proposition states that the graphs of HGNN, FastHyperGCN, and HyperGCN are the same irrespective of the signal values whenever the maximum size of a hyperedge is 3.

This explains why the three methods have comparable accuracies for Cora co-citaion and Citeseer co-citation hypergraphs. The mean hyperedge sizes are close to 3 (with comparitively lower deviations) as shown in Table 3. Hence the graphs of the three methods are more or less the same.

### Superior performance on Pubmed, DBLP, and Cora co-authorship

We see that HyperGCN performs statistically significantly (p-value of Welch t-test is less than 0.0001) compared to HGNN on the other three datasets. We believe this is due to large noisy hyperedges in real-world hypergraphs. An author can write papers from different topics in a co-authorship network or a paper typically cites papers of different topics in co-citation networks.

Average sizes in Table 3 show the presence of large hyperedges (note the large standard deviations). Clique expansion has edges on all pairs and hence potentially a larger number of hypernode pairs of different labels than the mediator graph of Figure 2, thus accumulating more noise.

### Preference of HyperGCN and FastHyperGCN over HGNN

To further illustrate superiority over HGNN on noisy hyperedges, we conducted experiments on synthetic hypergraphs each consisting of 1000 hypernodes, randomly sampled 500 hyperedges, and 2 classes with 500 hypernodes in each class. For each synthetic hypergraph, 100 hyperedges (each of size 5) were "pure", i.e., all hypernodes were from the same class while the other 400 hyperedges (each of size 20) contained hypernodes from both classes. The ratio, $\eta$, of hypernodes of one class to the other was varied from 0.75 (less noisy) to 0.50 (most noisy) in steps of 0.05.

Table 5 shows the results on synthetic data. We initialise features to random Gaussian of $d = 256$. We report mean error and deviation over 10 different synthetically generated hypergraphs. We see that for hyperedges with $\eta = 0.75, 0.7$ (mostly pure), HGNN is the superior model because it connects more similar vertices. However, as $\eta$ (noise) increases, our methods begin to outperform HGNN. Interestingly, for $\eta = 0.50$, FastHyperGCN even seems to outperform HyperGCN.

**Subset of DBLP**: We also trained all three models on a subset of DBLP (we call it sDBLP) by removing all hyperedges of size 2 and 3. The resulting hypergraph has around 8000 hyperedges with an average size of $8.5 \pm 8.8$. We report mean error over 10 different train-test splits in Table 5.

**Conclusion**: From the above analysis, we conclude that our proposed methods (HyperGCN and FastHyperGCN) should be preferred to HGNN for hypergraphs with large noisy hyperedges. This is also the case on experiments in combinatorial optimisation (Table 6) which we discuss next.

## 7   HyperGCN for combinatorial optimisation

Inspired by the recent sucesses of deep graph models as learning-based approaches for NP-hard problems [31, 38, 26, 19], we have used HyperGCN as a learning-based approach for the densest $k$-subhypergraph problem [15]. NP-hard problems on hypergraphs have recently been highlighted as crucial for real-world network analysis [2, 36]. Our problem is, given a hypergraph $(V, E)$, to find a subset $W \subseteq V$ of $k$ hypernodes so as to maximise the number of hyperedges contained in $V$, i.e., we wish to maximise the density given by $|e \in E : e \subseteq W|$.

A greedy heuristic for the problem is to select the $k$ hypernodes of the maximum degree. We call this "MaxDegree". Another greedy heuristic is to iteratively remove all hyperedges from the current (residual) hypergraph consisting of a hypernode of the minimum degree. We repeat the procedure $n-k$ times and consider the density of the remaining $k$ hypernodes. We call this "RemoveMinDegree".

**Experiments:** Table 6 shows the results. We trained all the learning-based models with a synthetically generated dataset. More details on the approach and the synthetic data are in the supplementary. As seen in Table 6, our proposed HyperGCN outperforms all the other approaches except for the pubmed dataset which contains a small number of vertices with large degrees and a large number of vertices with small degrees. The RemoveMinDegree baseline is able to recover all the hyperedges here.

**Visualisation**: Figure 3 shows the visualisations given by HGNN and HyperGCN on the Cora co-authorship clique-expanded hypergraph. We used Gephi's Force Atlas to space out the vertices. In general, a cluster of nearby vertices has multiple hyperedges connecting them. Clusters of only green vertices are ideal, this means the algorithm has likely included many hyperedges induced by the clusters. The figure of HyperGCN has more dense green clusters than that of HGNN.

Table 6: Results on the densest $k$-subhypergraph problem. We report density (higher is better) of the set of vertices obtained by each of the proposed approaches for $k = \frac{3|V|}{4}$. See section 7 for details.

| Dataset→ Approach↓ | Synthetic test set | DBLP co-authorship | Pubmed co-citation | Cora co-authorship | Cora co-citation | Citeseer co-citation |
|---|---|---|---|---|---|---|
| MaxDegree | $174 \pm 50$ | 4840 | 1306 | 194 | 544 | 507 |
| RemoveMinDegree | $147 \pm 48$ | **7714** | **7963** | 450 | 1369 | 843 |
| MLP | $174 \pm 56$ | 5580 | 1206 | 238 | 550 | 534 |
| MLP + HLR | $231 \pm 46$ | 5821 | 3462 | 297 | 952 | 764 |
| HGNN | $337 \pm 49$ | 6274 | 7865 | 437 | **1408** | **969** |
| 1-HyperGCN | $207 \pm 52$ | 5624 | 1761 | 251 | 563 | 509 |
| FastHyperGCN | $352 \pm 45$ | 7342 | 7893 | 452 | **1419** | **969** |
| HyperGCN | $\mathbf{359 \pm 49}$ | **7720** | **7928** | **504** | **1431** | **971** |
| **# hyperedges**, $|E|$ | 500 | 22535 | 7963 | 1072 | 1579 | 1079 |

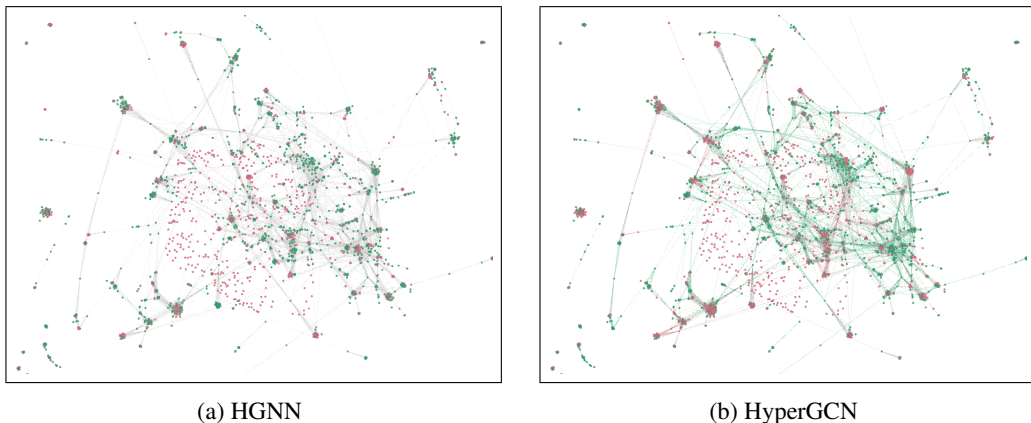

(a) HGNN            (b) HyperGCN

Figure 3: Green / pink hypernodes denote those the algorithm labels as positive / negative respectively.

## 8 Comparison of training times of FastHyperGCN and HGNN

We compared the average training times in Table 1. Both were run on a GeForce GTX 1080 Ti GPU machine. FastHyperGCN is faster because it uses a linear number of edges for each hyperedge while HGNN uses quadratic. It is also superior in terms of performance on hypergraphs with large noisy hyperedges (Table 5) and higly competitive on real-world data (Tables 4 and 6). Please see supplementary for the algorithms and time complexities of all the proposed methods and HGNN.

## 9 Conclusion

We have proposed HyperGCN, a new method of training GCN on hypergraph using tools from spectral theory of hypergraphs. We have shown HyperGCN's effectiveness in SSL and combinatorial optimisation. Approaches that assign importance to nodes [46, 35, 45] have improved results on SSL. HyperGCN may be augmented with such approaches for even more improved performance. One of the limitations of our approach is that the quality of the graph approximation obtained is highly dependent on the weight initialisation. We address this issue as part of future work.

## 10 Acknowledgement

Anand Louis was supported in part by SERB Award ECR/2017/003296 and a Pratiksha Trust Young Investigator Award. We acknowledge the support of Google India and NeurIPS in the form of an International Travel Grant, which enabled Naganand Yadati to attend the conference.

## Footnotes

*Work done while at Indian Institute of Science, Bangalore

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
