[Supplementary Material]

# Supplementary: Hypergraph convolutional network

## 1   Algorithms of our proposed methods

The forward propagation of a 2-layer graph convolutional network (GCN) [10] is

$$Z = \text{softmax}\left( \bar{A} \ \text{ReLU}\left( \bar{A} X \Theta^{(1)} \right) \Theta^{(2)} \right)$$

where $\bar{A} = \tilde{D}^{-\frac{1}{2}} \tilde{A} \tilde{D}^{-\frac{1}{2}}$, $\tilde{A} = A + I$, and $\tilde{D}_{ii} = \sum_{j=1}^{N} \tilde{A}_{ij}$ and $D = \text{diag}(d_1, \cdots, d_N)$ is the diagonal degree matrix with elements $d_i = \sum_{j=1, j \neq i}^{N} A_{ji}$. We provide algorithms for our three proposed methods:

- HyperGCN - Algorithm 1
- FastHyperGCN - Algorithm 2
- 1-HyperGCN - Algorithm 3

---

**Algorithm 1** Algorithm for HyperGCN

**Input**: An attributed hypergraph $\mathcal{H} = (V, E, X)$, with attributes $X$, a set of labelled vertices $\mathcal{V}_L$
**Output** All hypernodes in $V - \mathcal{V}_L$ labelled

1: **for** each epoch $\tau$ of training **do**
2:     **for** layer $l = 1, 2$ of the network **do**
3:         set $A_{vv}^{(l)} = 1$ For all hypernodes $v \in V$
4:         let $\Theta = \Theta^{\tau}$ be the parameters For the current epoch
5:         **for** $e \in E$ **do**
6:             $H \leftarrow$ hidden representation matrix of layer $l - 1$
7:             $i_e, j_e := \text{argmax}_{i,j \in e} ||H_i(\Theta^{(l)}) - H_j(\Theta^{(l)})||_2$
8:             $A_{i_e, j_e}^{(l)} = A_{j_e, i_e}^{(l)} = \frac{1}{2|e|-3}$
9:             $K_e := \{k \in e : \ k \neq i_e, k \neq j_e\}$
10:            **for** $k \in K_e$ **do**
11:                $A_{i_e, k}^{(l)} = A_{k, i_e}^{(l)} = \frac{1}{2|e|-3}$
12:                $A_{j_e, k}^{(l)} = A_{k, j_e}^{(l)} = \frac{1}{2|e|-3}$
13:            **end for**
14:         **end for**
15:     **end for**
16:     $Z = \text{softmax}\left( \bar{A}^{(2)} \ \text{ReLU}\left( \bar{A}^{(1)} X \Theta^{(1)} \right) \Theta^{(2)} \right)$
17:     update parameters $\Theta^{\tau}$ to minimise cross entropy loss on the set of labelled hypernodes $\mathcal{V}_L$
18: **end for**
19: label the hypernodes in $V - \mathcal{V}_L$ using $Z$

---

---

**Algorithm 2** Algorithm for FastHyperGCN

---

> **Input**: An attributed hypergraph $\mathcal{H} = (V, E, X)$, with attributes $X$, a set of labelled vertices $\mathcal{V}_L$
> **Output** All hypernodes in $V - \mathcal{V}_L$ labelled

set $A_{vv} = 1$ for all hypernodes $v \in V$
$i_e, j_e := \mathrm{argmax}_{i,j \in e} ||X_i - X_j||_2$
**for** $e \in E$ **do**
    $A_{i_e, j_e} = A_{j_e, i_e} = \frac{1}{2|e|-3}$
    $K_e := \{k \in e : \ k \neq i_e, k \neq j_e\}$
    **for** $k \in K_e$ **do**
        $A_{i_e, k} = A_{k, i_e} = \frac{1}{2|e|-3}$
        $A_{j_e, k} = A_{k, j_e} = \frac{1}{2|e|-3}$
    **end for**
**end for**
**for** each epoch $\tau$ of training **do**
    let $\Theta = \Theta^\tau$ be the parameters for the current epoch

$$Z = \mathrm{softmax}\left( \bar{A} \ \mathrm{ReLU}\left( \bar{A} X \Theta^{(1)} \right) \Theta^{(2)} \right)$$

    update parameters $\Theta^\tau$ to minimise cross entropy loss on the set of labelled hypernodes $\mathcal{V}_L$
**end for**
label the hypernodes in $V - \mathcal{V}_L$ using $Z$

---

---

**Algorithm 3** Algorithm for 1-HyperGCN

---

> **Input**: An attributed hypergraph $\mathcal{H} = (V, E, X)$, with attributes $X$, a set of labelled vertices $\mathcal{V}_L$
> **Output** All hypernodes in $V - \mathcal{V}_L$ labelled

**for** each epoch $\tau$ of training **do**
    **for** layer $l = 1, 2$ of the network **do**
        set $A_{vv}^{(l)} = 1$ for all hypernodes $v \in V$
        let $\Theta = \Theta^\tau$ be the parameters for the current epoch
        **for** $e \in E$ **do**
            $H \leftarrow$ hidden representation matrix of layer $l - 1$
            $i_e, j_e := \mathrm{argmax}_{i,j \in e} ||H_i(\Theta^{(l)}) - H_j(\Theta^{(l)})||_2$
            $A_{i_e, j_e}^{(l)} = A_{j_e, i_e}^{(l)} = \frac{1}{|e|}$
        **end for**
    **end for**

$$Z = \mathrm{softmax}\left( \bar{A}^{(2)} \ \mathrm{ReLU}\left( \bar{A}^{(1)} X \Theta^{(1)} \right) \Theta^{(2)} \right)$$

    update parameters $\Theta^\tau$ to minimise cross entropy loss on the set of labelled hypernodes $\mathcal{V}_L$
**end for**
label the hypernodes in $V - \mathcal{V}_L$ using $Z$

---

## 1.1   Time complexity

Given an attributed hypergraph $(V, E, X)$, let $d$ be the number of initial features, $h$ be the number of hidden units, and $l$ be the number of labels. Further, let $T$ be the total number of epochs of training. Define

$$N := \sum_{e \in E} |e|, \qquad N_m := \sum_{e \in E} \Big( 2|e| - 3 \Big), \qquad N_c := \sum_{e \in E} {}^{|e|}C_2$$

- HyperGCN takes $O\Big( T\Big( N + N_m h(d + c) \Big) \Big)$ time

- 1-HyperGCN takes $O\Big( TN\Big( 1 + h(d + c) \Big) \Big)$ time

17     • FastHyperGCN takes $O\Big(TN_m h(d+c)\Big)$ time

18     • HGNN takes $O\Big(TN_c h(d+c)\Big)$ time

# 2 HyperGCN for combinatorial optimisation

20 Inspired by the recent sucesses of deep graph models as learning-based approaches for NP-hard
21 problems [13, 15, 11, 7], we have used HyperGCN as a learning-based approach for the densest
22 $k$-subhypergraph problem [3], an NP-hard hypergraph problem. The problem is given a hypergraph
23 $(V, E)$, find a subset $W \subseteq V$ of $k$ hypernodes so as to maximise the number of hyperedges contained
24 in (induced by) $V$ i.e. we intend to maximise the density given by

$$|e \in E : e \subseteq W|$$

25 One natural greedy heuristic approach for the problem is to select the $k$ hypernodes of the maximum
26 degree. We call this approach "MaxDegree". Another greedy heuristic approach is to iteratively
27 remove all the hyperedges from the current (residual) hypergraph containing a hypernode of the
28 minimum degree. We repeat the procedure $n - k$ times and consider the density of the remaining $k$
29 hypernodes. We call this approach "RemoveMinDegree".

## 2.1 Our approach

31 A natural approach to the problem is to train HyperGCN to perform the labelling. In other words,
32 HyperGCN would take an input hypergraph $(V, E)$ as input and output a binary labelling of the
33 hypernodes $v \in V$. A natural output representation is a probability map in $[0, 1]^{|V|}$ that indicates how
34 likely each hypernode is to belong to $W$.

35 Let $\mathcal{D} = \{(V_i, E_i), l_i\}$ be a training set, where $(V_i, E_i)$ is an input hypergraph and $l_i \in \{0, 1\}^{|V| \times 1}$
36 is one of the optimal solutions for the NP-hard hypergraph problem. The HyperGCN model learns
37 its parameters $\Theta$ and is trained to predict $l_i$ given $(V_i, E_i)$. During training we minimise the binary
38 cross-entropy loss $L$ for each training sample $\{(V_i, E_i), l_i\}$ Additionally we generate $M$ different
39 probability maps to minimise the hindsight loss i.e. $\sum_i \min_m L^{(m)}$ where $L^{(m)}$ is the cross-entropy
40 loss corresponding to the $m$-th probability map. Generating multiple probability maps has the
41 advantage of generating diverse solutions [13].

## 2.2 Experiments: Training data

43 To generate a sample $\{(V, E), l\}$ in the training set $\mathcal{D}$, we fix a vertex set $W$ of $k$ vertices chosen
44 uniformly randomly. We generate each hyperedge $e \in E$ such that $e \subseteq W$ with high probability $p$.
45 Note that $e \subseteq V - W$ with probability $1 - p$. We give the algorithm to generate a sample $\{(V, E), l\}$.

---

**Algorithm 4** Algorithm for generating a training sample

     **Input**: A hypergraph $(V, E)$ and a dense set of vertices $W$ $\mathcal{V}_L$
     **Output** A hypergraph $(V, E)$ and a dense set of vertices $W$

   $|E| \leftarrow \frac{|V|}{2}$
   $W \leftarrow$ subset of $V$ of size $k$ chosen uniformly randomly
   **for** $i = 1, 2, \cdots, |E|$ **do**
     $|e| \sim \{2, 3, \cdots 10\}$ chosen uniformly randomly
     sample $e$ from $W$ with probability $p$
     sample $e$ from $V - W$ with probability $1 - p$
   **end for**

---

## 2.3 Experiments: Results

47 We generated 5000 training samples with the number of hypernodes $|V|$ uniformly randomly chosen
48 from $\{1000, 2000, \cdots, 5000\}$. We fix $|E| = \frac{|V|}{2}$ as this is mostly the case for real-world hypergraphs.

Table 1: Results on the densest $k$-subhypergraph problem. We report density (higher is better) of the set of vertices obtained by each of the proposed approaches for $k = \frac{3|V|}{4}$. See Section 2 for details.

| Dataset→ | Synthetic | DBLP | Pubmed | Cora | Cora | Citeseer |
| Approach↓ | test set | co-authorship | co-citation | co-authorship | co-citation | co-citation |
| --- | --- | --- | --- | --- | --- | --- |
| MaxDegree | $174 \pm 50$ | 4840 | 1306 | 194 | 544 | 507 |
| RemoveMinDegree | $147 \pm 48$ | **7714** | **7963** | 450 | 1369 | 843 |
| MLP | $174 \pm 56$ | 5580 | 1206 | 238 | 550 | 534 |
| MLP + HLR | $231 \pm 46$ | 5821 | 3462 | 297 | 952 | 764 |
| HGNN | $337 \pm 49$ | 6274 | 7865 | 437 | **1408** | **969** |
| 1-HyperGCN | $207 \pm 52$ | 5624 | 1761 | 251 | 563 | 509 |
| FastHyperGCN | $352 \pm 45$ | 7342 | 7893 | 452 | **1419** | **969** |
| HyperGCN | $\mathbf{359 \pm 49}$ | **7720** | **7928** | **504** | 1431 | **971** |
| **# hyperedges**, $|E|$ | 500 | 22535 | 7963 | 1072 | 1579 | 1079 |

(a) RemoveMinDegree       (b) HyperGCN

Figure 1: Green / pink hypernodes denote those the algorithm labels as positive / negative respectively.

Further we chose $e \in E$ such that $|e|$ is uniformly randomly chosen from $\{2, \cdots, 10\}$ as this is also mostly the case for real-world hypergraphs. We compared all our proposed approaches viz. 1-HyperGCN, HyperGCN, and FastHyperGCN against the baselines MLP, MLP+HLR and the state-of-the art HGNN. We also compared against the greedy heuristics MaxDegree and RemoveMinDegree. We train all the deep models using the same hyperparameters of [13] and report the results for $p = 0.75$ and $k = \frac{3|V|}{4}$ in Table 1. We test all the models on a synthetically generated test set of hypergraphs with 1000 vertices for each. We also test the models on the five real-world hypergraphs used for SSL experiments. As we can see in the table our proposed HyperGCN outperforms all the other approaches except for the pubmed dataset which contains a small number of vertices with large degrees and a large number of vertices with small degrees. The RemoveMinDegree baseline is able to recover all the hyperedges in the pubmed dataset. Moreover FastHyperGCN is competitive with HyperGCN as the number of hypergraphs in the training data is large.

## 2.4 Qualitative analysis

Figure 1 shows the visualisations given by RemoveMinDegree and HyperGCN on the Cora co-authorship hypergraph. We used Gephi's Force Atlas to space out the vertices. In general, a cluster of nearby vertices has multiple hyperedges connecting them. Clusters of only green vertices indicate the method has likely included all vertices within the hyperedges induced by the cluster. The figure of HyperGCN has more dense green clusters than that of RemoveMinDegree. Figure 2 shows the results of HGNN vs. HyperGCN.

(a) HGNN                                                       (b) HyperGCN

Figure 2: Green / pink hypernodes denote those the algorithm labels as positive / negative respectively.

# 3  Sources of the real-world datasets

**Co-authorship data**: All documents co-authored by an author are in one hyperedge. We used the author data[1] to get the co-authorship hypergraph for cora. We manually constructed the DBLP dataset from Arnetminer[2].

**Co-citation data**: All documents cited by a document are connected by a hyperedge. We used cora, citeseer, pubmed from [3] for co-citation relationships. We removed hyperedges which had exactly one hypernode as our focus in this work is on hyperedges with two or more hypernodes. Each hypernode (document) is represented by bag-of-words features (feature matrix $X$).

## 3.1  Construction of the DBLP dataset

We downloaded the entire dblp data from `https://aminer.org/lab-datasets/citation/DBLP-citation-Jan8.tar.bz`. The steps for constructing the dblp dataset used in the paper are as follows:

- We defined a set of 6 conference categories (classes for the SSL task) as "algorithms", "database", "programming", "datamining", "intelligence", and "vision"

- For a total of 4304 venues in the entire dblp dataset we took papers from only a subset of venues from `https://en.wikipedia.org/wiki/List_of_computer_science_conferences` corresponding to the above 6 conferences

- From the venues of the above 6 conference categories, we got 22535 authors publishing at least two documents for a total of 43413

- We took the abstracts of all these 43413 documents, constructed a dictionary of the most frequent words (words with frequency more than 100) and this gave us a dictionary size of 1425

# 4  Experiments on datasets with categorical attributes

We closely followed the experimental setup of the baseline model [19]. We experimented on three different datasets viz., mushroom, covertype45, and covertype67 from the UCI machine learning repository [5]. Properties of the datasets are summarised in Table 2. The task for each of the three datasets is to predict one of two labels (binary classification) for each unlabelled instance (hypernode). The datasets contain instances with categorical attributes. To construct the hypergraph, we treat each attribute value as a hyperedge, i.e., all instances (hypernodes) with the same attribute value are contained in a hyperedge. Because of this particular definition of a hyperedge clique expansion

Table 2: Summary of the three UCI datasets used in the experiments in Section 4

| property/dataset | mushroom | covertype45 | covertype67 |
|---|---|---|---|
| number of hypernodes, $|V|$ | 8124 | 12240 | 37877 |
| number of hyperedges, $|E|$ | 112 | 104 | 125 |
| number of edges in clique expansion | $65,999,376$ | $143,008,092$ | $1,348,219,153$ |
| number of classes, $q$ | 2 | 2 | 2 |

Figure 3: Test errors (lower is better) comparing HyperGCN_with_mediators with the non-neural baseline [19] on the UCI datasets. HyperGCN_with_mediators offers superior performance. Comparing against GCN on Clique Expansion is unfair. Please see below for details.

is destined to produce an almost fully connected graph and hence GCN on clique expansion will be unfair to compare against. Having shown that HyperGCN is superior to 1-HyperGCN in the relational experiments, we compare only the former and the non-neural baseline [19]. We have calledHyperGCN as HyperGCN_with_mediators. We used the incidence matrix (that encodes the hypergraph structure) as the data matrix $X$. We trained HyperGCN_with_mediators for the full 200 epochs and we used the same hyperparameters as in [10].

As in [19], we performed 100 trials for each $|V_L|$ and report the mean accuracy (averaged over the 100 trials). The results are shown in Figure 3. We find that HyperGCN_with_mediators model generally does better than the baselines. We believe that this is because of the powerful feature extraction capability of HyperGCN_with_mediators.

## 4.1 GCN on clique expansion

We reiterate that clique expansion, i.e., HGNN [6] for all the three datasets produce almost fuly connected graphs and hence clique expansion does not have any useful information. So, GCN on clique expansion is unfair to compare against (HGNN does not learn any useful weights for classification because of the fully connected nature of the graph).

## 4.2 Relevance of SSL

The main reason for performing these experiments, as pointed out in the publicly accessible NIPS reviews[4] of the total variation on hypergraphs [9], is to show that the proposed method (the primal-dual hybrid gradient method in their case and the HyperGCN_with_mediators method in our case) has improved results on SSL, even if SSL is not very relevant in the first place.

We do not claim that SSL with HyperGCN_with_mediators is the best way to go about handling these categorical data but we do claim that, *given this built hypergraph albeit from non-relational data*, it has superior results compared to the previous best non-neural hypergraph-based SSL method [19] in the literature and that is why we have followed their experimental setup.

## 5 Derivations

We show how the graph convolutional network (GCN) [10] has its roots from the convolution theorem [14].

Table 3: Results on ***Pubmed co-citation*** hypergraph. Mean test error $\pm$ standard deviation (lower is better) over 100 trials for different values of $|V_L|$. We randomly sampled the same number of labelled hypernodes from each class and hence we chose each $|V_L|$ to be divisible by $q$ with $\frac{|V_L|}{|V|}$ 0.2 to 1%.

| Available data | Method | 39 0.2% | 78 0.4% | 120 0.6% | 159 0.8% | 198 1% |
|---|---|---|---|---|---|---|
| $\mathcal{H}$ | CI | $62.61 \pm 1.69$ | $58.53 \pm 1.25$ | $55.71 \pm 1.03$ | $52.96 \pm 0.79$ | $50.21 \pm 0.56$ |
| $\mathbf{X}$ | MLP | $43.85 \pm 7.80$ | $35.17 \pm 4.92$ | $32.04 \pm 2.31$ | $30.70 \pm 1.61$ | $28.87 \pm 1.16$ |
| $\mathcal{H}, \mathbf{X}$ | MLP + HLR | $42.31 \pm 6.99$ | $33.69 \pm 4.49$ | $31.79 \pm 2.38$ | $30.18 \pm 1.54$ | $28.09 \pm 1.29$ |
| $\mathcal{H}, \mathbf{X}$ | HGNN | $37.99 \pm 6.45$ | $33.01 \pm 4.25$ | $31.14 \pm 2.23$ | $29.41 \pm 1.47$ | $26.96 \pm 1.35$ |
| $\mathcal{H}, \mathbf{X}$ | 1-HyperGCN | $43.62 \pm 7.18$ | $34.58 \pm 4.24$ | $31.88 \pm 2.78$ | $30.08 \pm 1.53$ | $28.90 \pm 1.29$ |
| $\mathcal{H}, \mathbf{X}$ | FastHyperGCN | $39.72 \pm 6.45$ | $32.67 \pm 3.91$ | $30.66 \pm 2.45$ | $29.48 \pm 1.60$ | $26.55 \pm 1.31$ |
| $\mathcal{H}, \mathbf{X}$ | HyperGCN | $\mathbf{33.33 \pm 7.01}$ | $\mathbf{31.71 \pm 4.37}$ | $\mathbf{28.84 \pm 2.60}$ | $\mathbf{25.56 \pm 1.55}$ | $\mathbf{23.97 \pm 1.24}$ |

## 5.1 Graph signal processing

We now briefly review essential concepts of graph signal processing that are important in the construction of ChebNet and graph convolutional networks. We need convolutions on graphs defined in the spectral domain. Similar to regular 1-D or 2-D signals, real-valued graph signals can be efficiently analysed via harmonic analysis and processed in the spectral domain [17]. To define spectral convolution, we note that the convolution theorem [14] generalises from classical discrete signal processing to take into account arbitrary graphs [16].

Informally, the *convolution theorem* says the convolution of two signals in one domain (say time domain) equals point-wise multiplication of the signals in the other domain (frequency domain). More formally, given a graph signal, $S : \mathcal{V} \to \mathbb{R}, S \in \mathbb{R}^N$, and a filter signal, $F : \mathcal{V} \to \mathbb{R}, F \in \mathbb{R}^N$, both of which are defined in the vertex domain (time domain), the convolution of the two signals, $C = S \star F$, satisfies

$$\hat{C} = \hat{S} \odot \hat{F} \tag{1}$$

where $\hat{S}$, $\hat{F}$, $\hat{C}$ are the graph signals in the spectral domain (frequency domain) corresponding, respectively, to $S$, $F$ and $S \star F$.

An essential operator for computing graph signals in the spectral domain is the symmetrically normalised graph Laplacian operator of $\mathcal{G}$, defined as

$$L = I - D^{-\frac{1}{2}} A D^{-\frac{1}{2}} \tag{2}$$

where $D = \text{diag}(d_1, \cdots, d_N)$ is the diagonal degree matrix with elements $d_i = \sum_{j=1, j \neq i}^{N} A_{ji}$. As the above graph Laplacian operator, $L$, is a real symmetric and positive semidefinite matrix, it admits spectral eigen decomposition of the form $L = U \Lambda U^T$, where, $U = [u_1, \cdots, u_N]$ forms an orthonormal basis of eigenvectors and $\Lambda = \text{diag}(\lambda_1, \cdots, \lambda_N)$ is the diagonal matrix of the corresponding eigenvalues with $0 = \lambda_1 \leq \cdots \leq \lambda_N \leq 2$.

The eigenvectors form a Fourier basis and the eigenvalues carry a notion of frequencies as in classical Fourier analysis. The graph Fourier transform of a graph signal $S = (S_1, \cdots, S_N) \in \mathbb{R}^N$, is thus defined as $\hat{S} = U^T S$ and the inverse graph Fourier transform turns out to be $S = U\hat{S}$, which is the same as,

$$S_i = \sum_{j=1}^{N} \hat{S}(\lambda_j) u_j(i) \quad \text{for } i \in \mathcal{V} = \{1, \cdots, N\} \tag{3}$$

The convolution theorem generalised to graph signals 1 can thus be rewritten as $U^T C = \hat{S} \odot \hat{F}$. It follows that $C = U(\hat{S} \odot \hat{F})$, which is the same as

$$C_i = \sum_{j=1}^{N} \hat{S}(\lambda_j) \hat{F}(\lambda_j) u_j(i) \quad \text{for } i \in \mathcal{V} = \{1, \cdots, N\} \tag{4}$$

Table 4: Results on ***DBLP co-authorship*** hypergraph. Mean test error $\pm$ standard deviation (lower is better) over 100 trials for different values of $|V_L|$. We randomly sampled the same number of labelled hypernodes from each class and hence we chose each $|V_L|$ to be divisible by $q$ with $\frac{|V_L|}{|V|}$ 1 to 5%.

| Available data | Method | 438 1% | 870 2% | 1302 3% | 1740 4% | 2172 5% |
|---|---|---|---|---|---|---|
| $\mathcal{H}$ | CI | $61.32 \pm 1.58$ | $59.39 \pm 1.37$ | $56.95 \pm 1.12$ | $54.81 \pm 0.94$ | $51.33 \pm 0.66$ |
| $\mathbf{X}$ | MLP | $44.57 \pm 7.19$ | $42.23 \pm 4.88$ | $38.89 \pm 3.62$ | $37.77 \pm 2.02$ | $35.12 \pm 1.57$ |
| $\mathcal{H}, \mathbf{X}$ | MLP + HLR | $34.54 \pm 7.49$ | $33.50 \pm 4.17$ | $32.77 \pm 3.16$ | $30.42 \pm 2.07$ | $29.21 \pm 1.94$ |
| $\mathcal{H}, \mathbf{X}$ | HGNN | $30.62 \pm 8.02$ | $27.09 \pm 4.48$ | $26.18 \pm 3.29$ | $25.65 \pm 2.08$ | $\mathbf{24.02 \pm 1.91}$ |
| $\mathcal{H}, \mathbf{X}$ | 1-HyperGCN | $40.17 \pm 6.99$ | $36.99 \pm 4.78$ | $34.44 \pm 3.43$ | $33.87 \pm 2.39$ | $32.11 \pm 1.96$ |
| $\mathcal{H}, \mathbf{X}$ | FastHyperGCN | $34.03 \pm 7.59$ | $29.93 \pm 4.35$ | $28.57 \pm 3.13$ | $27.34 \pm 2.06$ | $25.23 \pm 1.84$ |
| $\mathcal{H}, \mathbf{X}$ | HyperGCN | $\mathbf{28.51 \pm 7.73}$ | $\mathbf{25.45 \pm 4.32}$ | $\mathbf{24.69 \pm 3.08}$ | $\mathbf{24.09 \pm 2.02}$ | $23.96 \pm 1.98$ |

## 5.2 ChebNet convolution

We could use a non-parametric filter $\hat{F}(\lambda_j) = \theta_j$ for $j \in \{1, \cdots, N\}$ but there are two limitations: (i) they are not localised in space (ii) their learning complexity is $O(N)$. The two limitations above contrast with with traditional CNNs where the filters are localised in space and the learning complexity is independent of the input size. It is proposed by [4] to use a polynomial filter to overcome the limitations. A polynomial filter is defined as:

$$\hat{F}(\lambda_j) = \sum_{k=0}^{K} w_k \lambda_j^k \quad \text{for } j \in \{1, \cdots, N\} \tag{5}$$

Using 5 in 4, we get $C_i = \sum_{j=1}^{N} \hat{S}(\lambda_j) \left( \sum_{k=0}^{K} w_k \lambda_j^k \right) u_j(i)$ for $i \in \mathcal{V} = \{1, \cdots, N\}$. From the definition of an eigenvalue, we have $Lu_j = \lambda_j u_j$ and hence $L^k u_j = \lambda_j^k u_j$ for a positive integer $k$ and for $j \in \{1, \cdots, N\}$. Therefore,

$$
\begin{aligned}
C_i &= \sum_{j=1}^{N} \hat{S}(\lambda_j) \left( \sum_{k=0}^{K} w_k L_i^k \right) u_j(i) \\
&= \left( \sum_{k=0}^{K} w_k L_i^k \right) \sum_{j=1}^{N} \hat{S}(\lambda_j) u_j(i) \\
&= \left( \sum_{k=0}^{K} w_k L_i^k \right) S_i
\end{aligned}
\tag{6}
$$

Hence,

$$C = \left( \sum_{k=0}^{K} w_k L^k \right) S \tag{7}$$

The graph convolution provided by Eq. 7 uses the monomial basis $1, x, \cdots, x^K$ to learn filter weights. Monomial bases are not optimal for training and not stable under perturbations because they do not form an orthogonal basis. It is proposed by [4] to use the orthogonal Chebyshev polynomials [8] (and hence the name ChebNet) to recursively compute the powers of the graph Laplacian.

A Chebyshev polynomial $T_k(x)$ of order $k$ can be computed recursively by the stable recurrence relation $T_k(x) = 2xT_{k-1}(x) - T_{k-2}(x)$ with $T_0 = 1$ and $T_1 = x$. These polynomials form an orthogonal basis in $[-1, 1]$. Note that the eigenvalues of the symmetrically normalised graph Laplacian 2 lie in the range $[0, 2]$. Through appropriate scaling of eigenvalues from $[0, 2]$ to $[-1, 1]$

Table 5: Results on ***Cora co-authorship*** hypergraph. Mean test error $\pm$ standard deviation (lower is better) over 100 trials for different values of $|V_L|$. We randomly sampled the same number of labelled hypernodes from each class and hence we chose each $|V_L|$ to be divisible by $q$.

| Available data | Method | 42 | 98 | 140 | 203 |
|---|---|---|---|---|---|
| $\mathcal{H}$ | CI | $67.72 \pm 0.60$ | $58.55 \pm 0.53$ | $55.45 \pm 0.55$ | $51.44 \pm 0.32$ |
| $\mathbf{X}$ | MLP | $61.32 \pm 4.86$ | $47.69 \pm 2.36$ | $41.25 \pm 1.85$ | $37.76 \pm 1.32$ |
| $\mathcal{H}, \mathbf{X}$ | MLP + HLR | $54.31 \pm 5.12$ | $41.06 \pm 2.53$ | $34.87 \pm 1.78$ | $32.21 \pm 1.43$ |
| $\mathcal{H}, \mathbf{X}$ | HGNN | $45.23 \pm 5.03$ | $34.08 \pm 2.40$ | $31.90 \pm 1.87$ | $\mathbf{28.92 \pm 1.49}$ |
| $\mathcal{H}, \mathbf{X}$ | 1-HyperGCN | $50.26 \pm 4.78$ | $39.01 \pm 1.76$ | $36.22 \pm 2.21$ | $32.78 \pm 1.63$ |
| $\mathcal{H}, \mathbf{X}$ | HyperGCN | $\mathbf{43.86 \pm 4.78}$ | $\mathbf{33.83 \pm 1.81}$ | $\mathbf{30.08 \pm 1.80}$ | $29.08 \pm 1.44$ |

i.e. $\tilde{\lambda}_j = \frac{2\lambda_j}{\lambda_N} - 1$ for $j = \{1, \cdots, N\}$, where $\lambda_N$ is the largest eigenvalue, the filter in 5 can be parametrised as the truncated expansion

$$\hat{F}(\lambda_j) = \sum_{k=0}^{K} w_k T_k(\tilde{\lambda}_j) \quad \text{for } j \in \{1, \cdots, N\} \tag{8}$$

From Eq. 6, it follows that

$$C = \left( \sum_{k=0}^{K} w_k T_k(\tilde{L}) \right) S \quad \text{where } \tilde{L} = \frac{2L}{\lambda_N} - I \tag{9}$$

### 5.3 Graph convolutional network (GCN): first-order approximation of ChebNet

The spectral convolution of 9 is $K$-localised since it is a $K^{th}$-order polynomial in the Laplacian i.e. it depends only on nodes that are at most $K$ hops away. [10] simplify 9 to $K = 1$ i.e. they use simple filters operating on 1-hop neighbourhoods of the graph. More formally,

$$C = \left( w_0 + w_1 \tilde{L} \right) S \tag{10}$$

and also,

$$\hat{F}(\lambda_j) = w_0 + w_1 \tilde{\lambda}_j \quad \text{for } j \in \{1, \cdots, N\} \tag{11}$$

The main motivation here is that 10 is not limited to the explicit parameterisation given by the Chebyshev polynomials. Intuitively such a model cannot overfit on local neighbourhood structures for graphs with very wide node degree distributions, common in real-world graph datasets such as citation networks, social networks, and knowledge graphs.

In this formulation, [10] further approximate $\lambda_N \approx 2$, as the neural network parameters can adapt to the change in scale during training. To address overfitting issues and to minimise the number of matrix multiplications, they set $w_0 = -w_1 = \theta$. 10 now reduces to

$$C = \theta(I - \tilde{L})S = \theta(2I - L)S = \theta(I + D^{-\frac{1}{2}} A D^{-\frac{1}{2}})S \tag{12}$$

The filter parameter $\theta$ is shared over the whole graph and successive application of a filter of this form $K$ times then effectively convolves the $K^{th}$-order neighbourhood of a node, where $K$ is the number of convolutional layers (depth) of the neural network model. We note that the eigenvalues of $L$ are in $[0, 2]$ and hence the eigenvalues of $2I - L = I + D^{-\frac{1}{2}} A D^{-\frac{1}{2}}$ are also in the range $[0, 2]$. Repeated application of this operator can therefore lead to numerical instabilities and exploding/vanishing gradients. To alleviate this problem, a *renormalisation trick* can be used [10]:

$$I + D^{-\frac{1}{2}} A D^{-\frac{1}{2}} \rightarrow \tilde{D}^{-\frac{1}{2}} \tilde{A} \tilde{D}^{-\frac{1}{2}} \tag{13}$$

Table 6: Results on *Cora co-citation* hypergraph. Mean test error $\pm$ standard deviation (lower is better) over 100 trials for different values of $|V_L|$. We randomly sampled the same number of labelled hypernodes from each class and hence we chose each $|V_L|$ to be divisible by $q$.

| Available data | Method | 42 | 98 | 140 | 203 |
|---|---|---|---|---|---|
| $\mathcal{H}$ | CI | $79.25 \pm 1.34$ | $70.89 \pm 1.94$ | $64.40 \pm 0.81$ | $62.22 \pm 0.72$ |
| $\mathbf{X}$ | MLP | $63.31 \pm 5.23$ | $47.97 \pm 3.15$ | $42.14 \pm 1.78$ | $40.05 \pm 1.58$ |
| $\mathcal{H}, \mathbf{X}$ | MLP + HLR | $56.21 \pm 5.65$ | $43.32 \pm 3.27$ | $36.98 \pm 1.83$ | $33.88 \pm 1.46$ |
| $\mathcal{H}, \mathbf{X}$ | HGNN | $50.39 \pm 5.42$ | $\mathbf{35.62 \pm 3.11}$ | $\mathbf{32.41 \pm 1.82}$ | $\mathbf{29.78 \pm 1.55}$ |
| $\mathcal{H}, \mathbf{X}$ | 1-HyperGCN | $50.39 \pm 5.41$ | $38.01 \pm 3.12$ | $34.45 \pm 2.05$ | $31.67 \pm 1.57$ |
| $\mathcal{H}, \mathbf{X}$ | HyperGCN | $\mathbf{47.00 \pm 5.32}$ | $\mathbf{35.76 \pm 2.60}$ | $32.37 \pm 1.71$ | $29.98 \pm 1.45$ |

with $\tilde{A} = A + I$ and $\tilde{D}_{ii} = \sum_{j=1}^{N} \tilde{A}_{ij}$. Generalising the above to $p$ signals contained in the matrix $X \in \mathbb{R}^{N \times p}$ (also called the data matrix), and $r$ filter maps contained in the matrix $\Theta \in \mathbb{R}^{p \times r}$, the output convolved signal matrix will be:

$$\bar{A}X\Theta \quad \text{where } \bar{A} = \tilde{D}^{-\frac{1}{2}} \tilde{A} \tilde{D}^{-\frac{1}{2}} \tag{14}$$

## 5.4 GCNs for graph-based semi-supervised node classification

The GCN is conditioned on both the adjacency matrix $A$ (underlying graph structure) and the data matrix $X$ (input features). This allows us to relax certain assumptions typically made in graph-based SSL, for example, the cluster assumption [2] made by the explicit Laplacian-based regularisation methods. This setting is especially powerful in scenarios where the adjacency matrix contains information not present in the data (such as citation links between documents in a citation network or relations in a knowledge graph). The forward model for a simple two-layer GCN takes the following simple form:

$$Z = f_{GCN}(X, A) = \text{softmax}\left( \bar{A} \ \text{ReLU}\left( \bar{A}X\Theta^{(0)} \right) \Theta^{(1)} \right) \tag{15}$$

where $\Theta^{(0)} \in \mathbb{R}^{p \times h}$ is an input-to-hidden weight matrix for a hidden layer with $h$ hidden units and $\Theta^{(1)} \in \mathbb{R}^{h \times r}$ is a hidden-to-output weight matrix. The softmax activation function defined as $\text{softmax}(x_i) = \frac{\exp(x_i)}{\sum_i \exp(x_i)}$ is applied row-wise.

**Training** For semi-supervised multi-class classification with $q$ classes, we then evaluate the cross-entropy error over all the set of labelled examples, $\mathcal{V}_L$:

$$\mathcal{L} = -\sum_{i \in \mathcal{V}_L} \sum_{j=1}^{q} Y_{ij} \ln Z_{ij} \tag{16}$$

The weights of the graph convolutional network, viz. $\Theta^{(0)}$ and $\Theta^{(1)}$, are trained using gradient descent. Using efficient sparse-dense matrix multiplications for computing, the computational complexity of evaluating Eq. 15 is $O(|\mathcal{E}|phr)$ which is linear in the number of graph edges.

## 5.5 GCN as a special form of Laplacian smoothing

GCNs can be interpreted as a special form of symmetric Laplacian smoothing [12]. The Laplacian smoothing [18] on each of the $p$ input channels in the input feature matrix $X \in \mathbb{R}^{N \times p}$ is defined as:

$$\chi_i = (1 - \gamma)x_i + \gamma \sum_j \frac{\tilde{A}_{ij}}{d_i} x_j \quad i = 1, \cdots, N \tag{17}$$

Table 7: Results on *Citeseer co-citation* hypergraph. Mean test error $\pm$ standard deviation (lower is better) over 100 trials for different values of $|V_L|$. We randomly sampled the same number of labelled hypernodes from each class and hence we chose each $|V_L|$ to be divisible by $q$.

| Available data | Method | 42 | 102 | 138 | 198 |
|---|---|---|---|---|---|
| $\mathcal{H}$ | CI | $74.68 \pm 1.02$ | $71.90 \pm 0.82$ | $70.37 \pm 0.29$ | $68.84 \pm 0.24$ |
| $\mathbf{X}$ | MLP | $57.14 \pm 4.87$ | $45.80 \pm 2.43$ | $41.12 \pm 1.65$ | $39.09 \pm 1.32$ |
| $\mathcal{H}, \mathbf{X}$ | MLP + HLR | $53.21 \pm 4.65$ | $43.21 \pm 2.35$ | $37.75 \pm 1.59$ | $36.01 \pm 1.29$ |
| $\mathcal{H}, \mathbf{X}$ | HGNN | $\mathbf{50.75 \pm 4.73}$ | $\mathbf{39.67 \pm 2.21}$ | $\mathbf{37.40 \pm 1.61}$ | $\mathbf{35.20 \pm 1.35}$ |
| $\mathcal{H}, \mathbf{X}$ | 1-HyperGCN | $52.48 \pm 5.43$ | $41.26 \pm 2.54$ | $38.87 \pm 1.93$ | $36.46 \pm 1.46$ |
| $\mathcal{H}, \mathbf{X}$ | HyperGCN | $\mathbf{50.39 \pm 5.13}$ | $\mathbf{39.68 \pm 2.27}$ | $\mathbf{37.35 \pm 1.62}$ | $\mathbf{35.40 \pm 1.22}$ |

here $\tilde{A} = A + I$ and $d_i$ is the degree of node $i$. Equivalently the Laplacian smoothing can be written as $\chi = X - \gamma \tilde{D}^{-1}\tilde{L}X = (I - \gamma\tilde{D}^{-1}\tilde{L})X$ where $\tilde{L} = \tilde{D} - \tilde{A}$. Here $0 \le \gamma \le 1$ is a parameter which controls the weighting between the feature of the current vertex and those of its neighbours. If we let $\gamma = 1$, and replace the normalised Laplacian $\tilde{D}^{-1}\tilde{L}$ by the symmetrically normalised Laplacian $\tilde{D}^{-\frac{1}{2}}\tilde{L}\tilde{D}^{-\frac{1}{2}}$, then $\chi = (I - \tilde{D}^{-\frac{1}{2}}\tilde{L}\tilde{D}^{-\frac{1}{2}})X = \bar{A}X$, the same as in the expression 14.

Hence the graph convolution in the GCN is a special form of (symmetric) Laplacian smoothing. The Laplacian smoothing of Eq. 17 computes the new features of a node as the weighted average of itself and its neighbours. Since nodes in the same cluster tend to be densely connected, the smoothing makes their features similar, which makes the subsequent classification task much easier. Repeated application of Laplacian smoothing many times over leads to over-smoothing - the node features within each connected component of the graph will converge to the same values [12].

# 6 Hyperparameters and more experiments on SSL

Please see tables 3, 4, 5, 6, and 7 for the results on all the real-world hypergraph datasets.

Following a prior work [10], we used the following hyperparameters for all the models:

- hidden layer size: 32
- dropout rate: 0.5
- learning rate: 0.01
- weight decay: 0.0005
- number of training epochs: 200
- $\lambda$ for explicit Laplacian regularisation: 0.001

The Laplacian with mediators [1] allows a general set fo weights. We tried one other set of weights which assigns uniform weights on the edges to the mediators but zero weight on the edge between the maximally disparate vertices. On the sDBLP dataset, this approach achieves an accuracy of $41.71 \pm 2.9$. HyperGCN achieves $41.64 \pm 2.6$ and FastHyperGCN achieves $41.79 \pm 2.8$.

## Footnotes

[1]https://people.cs.umass.edu/ mccallum/data.html

[2]https://aminer.org/lab-datasets/citation/DBLP-citation-Jan8.tar.bz

[3]https://linqs.soe.ucsc.edu/data

[4]https://papers.nips.cc/paper/4914-the-total-variation-on-hypergraphs-learning-on-hypergraphs-revisited