[Reviews · NeurIPS 2019]

Reviewer 1



The relationships of many real-world networks are complex and go beyond pairwise associations. Hypergraphs provide a flexible and natural modeling tool to model such complex relationships. The authors propose HyperGCN, a novel way of training a GCN for semi-supervised learning on hypergraphs using tools from spectral theory of hypergraphs and introduce FastHyperGCN. They conduct some experiments on co-authorship and co-citation hypergraphs to demonstrate the effectiveness of HyperGCN, and provide theoretical analyses for the results. Strength: 1. The paper proposes 1-HyperGCN and HyperGCN using the hypergraph Laplacian and the generalized hypergraph Laplacian with mediators. FastHyperGCN is proposed for fast training, which computes the hypergraph Laplacian matrix only once before training. 2. HyperGCN is applied to combinatorial optimization such as densest k-subhypergraph problem. The paper is well presented and easy to follow, but it can be further improved by addressing the following potential issues: 1. The experimental setting is a little unclear. The training/test split ratio of the dataset is not reported in the paper. 2. HyperGCN should be compared with GCN models since GCN models are also designed for the semi-supervised task and can be easily applied to hypergraphs. 3. There are a few typos in the paper such as “Cora co-citaion” in line 203.

Reviewer 2



As the contributions detailed above, this paper has good originality, which nicely connects the recent developments of graph neural networks and spectral hypergraph theory. Moreover, extensive experiments in both semi-supervised learning and combinatorial optimization demonstrate the effectiveness of such a new connection. So I also think this paper has good quality and significance. The only issue I suspect is about clarity. As I am rather familiar with spectral hypergraph theory and kind of understand graph neural networks, this content reads acceptable for me. However, I do not think the review of new hypergraph Laplacian operators is adequate, especially for the NeurIPS community to better capture the idea. Along with the review of GCN, More background on hypergraph Laplacian operators should be introduced. --- I have read through the response. The authors have solved my questions. In the final version, please provide a better exposition of the new Laplacians that motivate your work and also additional evaluations on mediators.

Reviewer 3



This paper extends the previous GCNs to handle the hypergraph by proposing a HyperGCN variants. Although there are many works have done the similar effort, this paper reduces the computational complexity to the linear scale via a Hypergraph Laplacian operator. However, the main concern is that other advantages of the proposed method in this paper is not clear except the reduce of the computational complexity. This confuses the reviewer why the performance of HyperGCN outperforms previous Hypergraph neural networks since they use more complete information. Besides, the extension of HyperGCNs with mediators and the Fast version are incremental and straightforward without any effort based on previous works. The authors have not given some useful insight based on the extensions. Finally, the experiments lack of thorough comparison more recent Hypergraph models.

[Author Response · NeurIPS 2019]

We thank the reviewers for the reviews. We are glad to see that the reviewers liked the significant and novel contributions of the paper. We address specific questions and misunderstandings in the reviews.

**Reviewer #1**

**On train-test split ratio:** We refer the reviewer to *label rate* on Table 3 in our paper. These numbers are precisely the $\frac{train}{train+test}$ split ratios on each dataset.

**On a robust experimental setting:** As mentioned in the paper, we have reported mean accuracy and standard deviation over 100 different train-test splits in semi-supervised learning experiments to ensure a robust experimental setting. We refer the reviewer to the supplementary material/appendix for more experiments on different train-test split ratios. Moreover, as confirmed by Reviewer #2, extensive experiments in both semi-supervised learning and combinatorial optimisation demonstrate the effectiveness of our methods.

**On comparison against GCN:** We have thoroughly compared against HGNN [3] in all our experiments. HGNN uses the clique expansion [4]. Hence, it is exactly the GCN baseline that is easily extended to hypergraphs [1].

**On typos:** Thanks for pointing the typos out. We will fix them in the final version.

**Reviewer #2**

**On the lack of clarity:** Many thanks to the reviewer for pointing out the lack of clarity in the paper. We will include background reviews of all the new hypergraph Laplacian operators in the final version of the paper by providing a crisp overview for each work. We will ensure that the paper is self-contained so that the broader NeurIPS community can appreciate the significance of our work better.

**On 1-HyperGCN vs. FastHyperGCN:** It is to be noted that 1-HyperGCN samples one edge (in each epoch) while FastHyperGCN does use the mediators and hence samples $2|e| - 3$ edges for each hyperedge $e \in E$. Hence, it is quite intuitive that FastHyperGCN is superior to 1-HyperGCN.

**On different weights on edges:** We observed that uniform weights on all edges do give the best results compared to other weights (e.g. zero weight for $\{i_e, j_e\}$ and uniform weights for the remaining edges to the mediators). As suggested by the reviewer, we will include a comparison table in the appendix.

**On small issues:** Thanks for pointing these out. We will fix them in the final version.

**Reviewer #3**

**On insights into the proposed method:** We refer the reviewer to Section 6 of our paper for insights into the method. As confirmed by Reviewer #1, we have provided theoretical analyses on the results for insights.

**On when and why HyperGCN outperforms HGNN:** We refer the reviewer to Table 5 in our paper which provides insights into when HyperGCN outperforms HGNN [3]. Our methods produce sparser approximations which accumulate less noise and hence are superior on nosiy datasets as shown in Table 5.

**On comparison against more recent hypergraph methods:** We have thoroughly compared against the two known state-of-the-art semi-supervised learning methods on undirected hypergraphs viz., HGNN [3] and explicit Laplacian regularisation [2] (MLP + HLR in our paper).

# References

[1] Sameer Agarwal, Kristin Branson, and Serge Belongie. Higher order learning with graphs. In *ICML*, 2006.

[2] T.-H. Hubert Chan and Zhibin Liang. Generalizing the hypergraph laplacian via a diffusion process with mediators. In *COCOON*, 2018.

[3] Yifan Feng, Haoxuan You, Zizhao Zhang, Rongrong Ji, and Yue Gao. Hypergraph neural networks. In *AAAI*, 2019.

[4] Denny Zhou, Jiayuan Huang, and Bernhard Schölkopf. Learning with hypergraphs: Clustering, classification, and embedding. In *NIPS*, 2007.


[Meta-Review · NeurIPS 2019]

This submission provides a technique to more faithfully generalize graph convolutional networks to handle hypergraph data. The paper is original and makes nice connections to spectral hypergraph theory and there is demonstrated performance improvement over graph-based hypergraph modeling techniques like clique expansion. For these reasons, I am recommending to accept this paper. There is one specific issues that should be addressed in a camera version. Please add more discussion on why FastHyperGCN can be better than HyperGCN in practice (Table 5). The fast version is omitting information that would presumably be useful. The presentation here should be clarified. Is this just due to variance in the experiments? Additional clarification on the performance of 1-HyperGCN compared to FastHyperGCN would also be useful, as the reviewers still did not completely understand the author feedback on that point.